# Evolutionary dynamics and genomic features of the *Elizabethkingia anophelis* 2015 to 2016 Wisconsin outbreak strain

Amandine Perrin[1,2,3,*], Elise Larsonneur[1,2,4,*], Ainsley C. Nicholson[5,*], David J. Edwards[6,7], Kristin M. Gundlach[8], Anne M. Whitney[5], Christopher A. Gulvik[5], Melissa E. Bell[5], Olaya Rendueles[1,2], Jean Cury[1,2], Perrine Hugon[1,2], Dominique Clermont[9], Vincent Enouf[10], Vladimir Loparev[11], Phalasy Juieng[11], Timothy Monson[8], David Warshauer[8], Lina I. Elbadawi[12,13], Maroya Spalding Walters[14], Matthew B. Crist[14], Judith Noble-Wang[14], Gwen Borlaug[13], Eduardo P.C. Rocha[1,2], Alexis Criscuolo[3], Marie Touchon[1,2], Jeffrey P. Davis[13], Kathryn E. Holt[6,7], John R. McQuiston[5] & Sylvain Brisse[1,2,15]

An atypically large outbreak of *Elizabethkingia anophelis* infections occurred in Wisconsin. Here we show that it was caused by a single strain with thirteen characteristic genomic regions. Strikingly, the outbreak isolates show an accelerated evolutionary rate and an atypical mutational spectrum. Six phylogenetic sub-clusters with distinctive temporal and geographic dynamics are revealed, and their last common ancestor existed approximately one year before the first recognized human infection. Unlike other *E. anophelis*, the outbreak strain had a disrupted DNA repair *mutY* gene caused by insertion of an integrative and conjugative element. This genomic change probably contributed to the high evolutionary rate of the outbreak strain and may have increased its adaptability, as many mutations in protein-coding genes occurred during the outbreak. This unique discovery of an outbreak caused by a naturally occurring mutator bacterial pathogen provides a dramatic example of the potential impact of pathogen evolutionary dynamics on infectious disease epidemiology.

[1] Institut Pasteur, Microbial Evolutionary Genomics, F-75724 Paris, France. [2] CNRS, UMR 3525, F-75724 Paris, France. [3] Institut Pasteur, Hub Bioinformatique et Biostatistique, C3BI, USR 3756 IP CNRS, F-75724 Paris, France. [4] CNRS, UMS 3601 IFB-Core, F- 91198 Gif-sur-Yvette, France. [5] Special Bacteriology Reference Laboratory, Bacterial Special Pathogens Branch, Division of High Consequence Pathogens and Pathology, Centers for Disease Control and Prevention, Atlanta, Georgia 30329, USA. [6] Centre for Systems Genomics, University of Melbourne, Parkville, Victoria 3010, Australia. [7] Department of Biochemistry and Molecular Biology, Bio21 Molecular Science and Biotechnology Institute, University of Melbourne, Parkville, Victoria 3010, Australia. [8] Wisconsin State Laboratory of Hygiene, Madison, Wisconsin 53718, USA. [9] CIP—Collection de l'Institut Pasteur, Institut Pasteur, F-75724 Paris, France. [10] Institut Pasteur, Pasteur International Bioresources network (PIBnet), Plateforme de Microbiologie Mutualisée (P2M), F-75724 Paris, France. [11] Division of Scientific Resources, Centers for Disease Control and Prevention, Atlanta, Georgia 30329, USA. [12] Epidemic Intelligence Service, Centers for Disease Control and Prevention, Atlanta, Georgia 30329, USA. [13] Division of Public Health, Wisconsin Department of Health Services, Madison, Wisconsin 53701, USA. [14] Division of Healthcare Quality Promotion, Centers for Disease Control and Prevention, Atlanta, Georgia 30329, USA. [15] Institut Pasteur, Molecular Prevention and Therapy of Human Diseases, F-75724 Paris, France. * These authors contributed equally to this work. Correspondence and requests for materials should be addressed to J.R.M. (email: zje8@cdc.gov) or to S.B. (email: sylvain.brisse@pasteur.fr).

An outbreak of 66 laboratory-confirmed infections caused by the bacterial pathogen *Elizabethkingia anophelis* occurred in 2015–2016 in the USA states of Wisconsin (63 patients), Illinois (2 patients) and Michigan (1 patient). This was the largest ever documented *Elizabethkingia* outbreak, and the only one with illness onsets occurring primarily (89% of Wisconsin patients) in community settings. Isolates obtained from patients shared a unique genotype as defined by pulsed field gel electrophoresis, and the localized distribution of early cases was suggestive of a point source. A joint investigation by the Wisconsin Division of Public Health, Wisconsin State Laboratory of Hygiene and the Centers for Disease Control and Prevention (CDC) assessed many potential sources of the outbreak, including health-care products, personal care products, food, tap water and person-to-person transmission. The outbreak appeared to wane by mid-May 2016, and a source of infection had not yet been identified by September 2016. The ongoing investigation and updates on this outbreak are described by Centers for Disease Control and Prevention (https://www.cdc.gov/elizabethkingia/outbreaks/) and Wisconsin Department of Health Services (https://www.dhs.wisconsin.gov/disease/elizabethkingia.htm).

*E. anophelis* is a recently recognized species[1]. Despite recent genomic and experimental work[2–6], virulence factors or mechanisms of pathogenesis by *E. anophelis* are yet to be discovered. Knowledge of the ecology and epidemiology of this emerging pathogen is also in its infancy. All previously reported *Elizabethkingia* outbreaks have been health-care associated[7–9] although sporadic, community-acquired cases have been occasionally reported[10], as has a single instance of transmission of *E. anophelis* from mother to infant at birth[11]. Human infections have varied presentations, including meningitis and septicaemia[12–15]. Strains have been isolated from diverse environments such as hospital sinks (*E. meningoseptica* and *E. anophelis*)[6,7], the mosquito mid-gut (*E. anophelis*)[1] and the space station Mir (*E. miricola*)[16]. Therefore, *Elizabethkingiae* are generally regarded as environmental, and although *E. anophelis* has been recovered from the mid-gut of wild-caught *Anopheles* and *Aedes* mosquitoes[1], there is no indication that mosquitoes serve as a vector to transmit the bacteria to humans. *E. anophelis* is naturally resistant to multiple antimicrobial agents and harbours several genetic determinants of antimicrobial resistance, including multiple beta-lactamases and efflux systems[2,4,6,17,18]. *Elizabethkingia* species are phenotypically very similar, leading to misidentifications that compromise our understanding of the relative clinical importance of each species. Previously reported *E. meningoseptica* outbreaks may in fact have been caused by *E. anophelis,* as this latter species was recently reported to be the primary cause of clinically significant *Elizabethkingia* infections in Singapore[15].

The unique magnitude and setting of the Wisconsin outbreak and its elusive source prompted us to explore the genomic features of the outbreak strain, and compare them to other *Elizabethkingia* strains. We found that the outbreak strain represents a novel phylogenetic sublineage of *E. anophelis* and has unique genomic regions. Furthermore, it displayed exceptional evolutionary dynamism during the outbreak, likely caused by the insertion of the mobile integrative and conjugative element (ICE*Ea*1) into the *mutY* DNA repair gene.

## Results

### The outbreak is caused by a novel *E. anophelis* sublineage.

A phylogenetic analysis was performed with the 69 Wisconsin outbreak isolates (from 59 patients) and 45 comparative strains of *E. anophelis* and other *Elizabethkingia* species (Supplementary Fig. 1a). The tree revealed three major branches, each containing one of the three *Elizabethkingia* species (*E. meningoseptica, E. miricola* and *E. anophelis*). The *E. miricola* branch was the most heterogeneous and comprised, in addition to *E. miricola* strains, reference strains of the distinct genomospecies defined by DNA–DNA hybridization[19]: G4071 (genomospecies 2), G4075 (genomospecies 3) and G4122 (genomospecies 4). We, therefore, labelled this branch, which may comprise several species, as the *E. miricola* cluster. The type strain JM-87[T] of *E. endophytica* was placed within the *E. anophelis* branch, consistent with a recent report[20]. Eight clinical strains initially identified as *E. meningoseptica* were in fact members of the *E. anophelis* species. Additional discordances found between the phylogenetic position of several strains and their initial taxonomic designation (Supplementary Data 1) underscore the uncertainty associated with species determination for *Elizabethkingia* isolates[20].

The outbreak isolates made up a compact phylogenetic group within *E. anophelis* (sublineage 15 in Supplementary Fig. 1b), indicating that the outbreak was caused by a single ancestral strain. The long branch that separated the outbreak strain from all other sequenced *E. anophelis* strains showed that the outbreak strain is derived from a unique sublineage of *E. anophelis* that had not been previously described. The other *E. anophelis* strains were highly diverse, forming 14 other sublineages. Strains CIP 79.29 and GTC 09686 (sublineage 14) were most closely related to the outbreak strain but had a nucleotide divergence of ∼1%. These results show that the currently known sublineages of *E. anophelis* are not close relatives of the outbreak strain.

### Phylogenetic diversity and temporal and geographic dynamics.

Phylogenetic analysis of the Wisconsin isolates disclosed a highly dynamic outbreak, with a conspicuous genetic diversification into several sub-clusters (Fig. 1, Supplementary Fig. 2). Except for three outliers, all outbreak isolates derived from a single ancestor (node I, Fig. 1). We defined six main sub-clusters (sc1 to sc6, Fig. 1) based on visual inspection of the tree. Whereas sc1 branched off early, sub-clusters sc2 to sc6 shared a common ancestor (node II, Fig. 1).

Several patients were sampled on multiple occasions from 1 to 21 days apart, and from up to four different sites per patient. The cgMLST (core genome multilocus sequence typing) loci of groups of isolates from single patients were always identical, except for one single-nucleotide polymorphism (SNP) observed between isolate CSID 3000515962 and the three other isolates from the same patient. These results indicate that the pathogen population that infected each individual patient was dominated by a single genetic variant. In addition, these results underline the high reproducibility of the sequencing and genotyping processes.

The phylogenetic diversity within the outbreak clade provides an opportunity to estimate the temporal dynamics of the diversification of the outbreak strain. We first tested whether there was a temporal signal, that is, whether the root-to-tip distance was correlated with the date of sampling of bacterial isolates. Bayesian analysis with BEAST using randomized tip dates demonstrated a significant temporal signature (Supplementary Fig. 3), implying that the outbreak strain continued diversifying in a measurable way over the course of the outbreak. We next estimated a mean evolutionary rate of $5.98 \times 10^{-6}$ nucleotide substitutions per site per year (95% HPD (highest posterior density) = 3.47, 8.61) based on cgMLST genes, corresponding to 24 substitutions per genome per year. This analysis placed Node I, from which all but three (including the hypermutator, see below) infectious isolates were derived, at around July 2015, and the last common ancestor of all outbreak isolates at the end of December 2014 (95% HPD = January 2014, July 2015) (Supplementary Fig. 4). Using an independent whole-genome SNP approach, the evolutionary rate estimate was

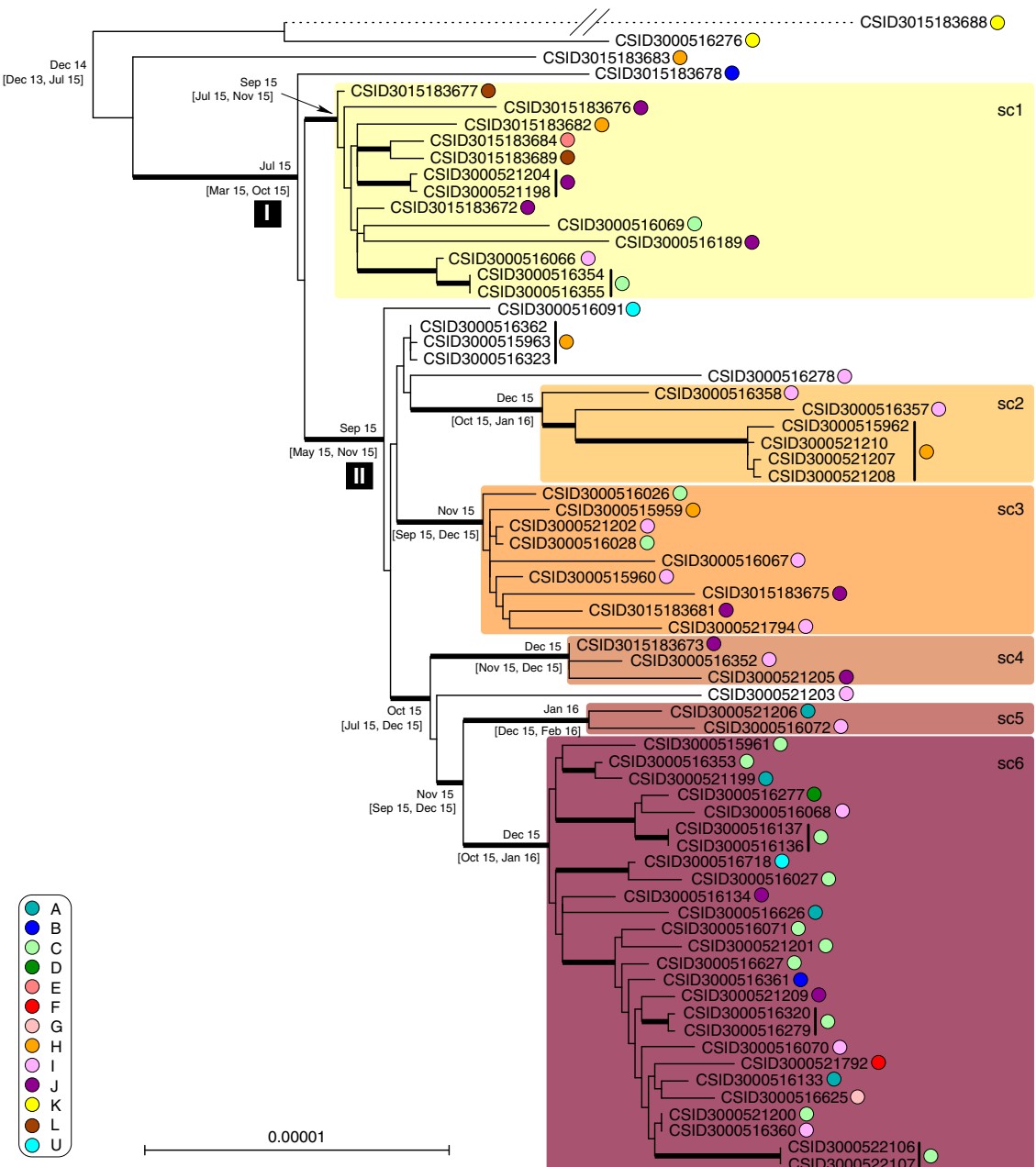

**Figure 1 | Phylogenetic tree of the outbreak isolates.** Maximum likelihood phylogenetic tree inferred from 3,411,033 aligned nucleotide characters (1,137,011 codons) based on cgMLST data. The tree was rooted based on phylogenetic analyses using epidemiologically unrelated *E. anophelis* strains as an outgroup. Thick branches have bootstrap support > 80% (200 replicates). The scale bar represents substitutions per site. Sub-clusters (sc) 1 to 6 are represented by coloured boxes. Counties A to L (and U for 'unspecified', attributed to the strains from outside of Wisconsin) are represented by coloured circles (see key on the left). Sets of isolates gathered from the same patient are indicated with vertical black lines after the isolate codes. Median Bayesian estimates of the month and year are provided for major internal branches (with 95% HPDs in square brackets). The branching position of the *mutS* isolate CSID 3015183688, denoted by the dashed branch line, was defined based on a separate analysis (using the same methods) and its branch length was divided by 5 for practical reasons.

$6.35 \times 10^{-6}$ nucleotide substitutions per site per year (95% HPD = 3.66, 9.07), and the date of the last common ancestor was estimated at August 2014 (95% HPD = June 2013, June 2015). These two approaches thus provided concordant results and suggested that the initial diversification of the outbreak strain predates the first identified human infection in this outbreak by approximately one year. Because the retrospective epidemiological analysis demonstrates that human cases of *E. anophelis* infection were likely not missed, these results suggest that the strain evolved in its reservoir during an approximately one-year

interval before contaminating the source of infection, and that further diversification occurred, either in the reservoir or in the source of infection, as the outbreak was ongoing.

Phylogenetic diversification followed both temporal and geographic trends (Fig. 2). Sub-cluster sc1 appeared first, in multiple locations during the first week, and was later supplemented by the other clusters, with an initial south-east drift of cases during the first 6 weeks. Sc6 appeared later and became the most common of the sub-clusters after February 1, coinciding with concentration of cases in the south-eastern-most

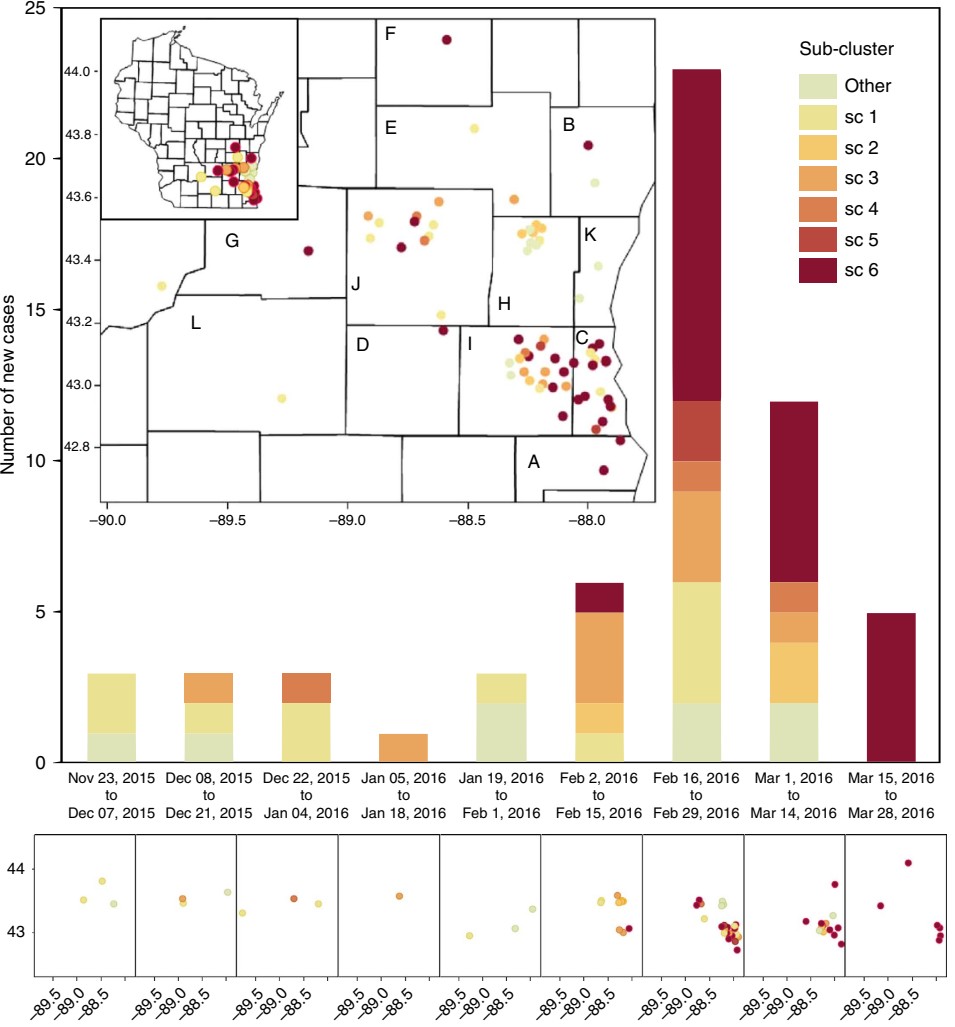

**Figure 2 | Temporal-spatial distribution of cases by genetic sub-cluster.** Case counts ($n=59$, over the three-state area) are presented in two-week intervals, as indicated below the histogram bars, based on the date of initial positive culture. Genetic sub-cluster colours (see key) correspond to those in the phylogenetic figures. Geographic distribution of Wisconsin cases ($n=56$) is displayed, overall (insets) and by two-week intervals (lower panel). The numbers along the $x$ and $y$ axis of the maps are longitude and latitude, respectively. Letters inside counties correspond to letters on the lower left key on Fig. 1.

corner of the 12 county outbreak region during the outbreak peak and followed by a wider geographic spread of sc6 after March 1. This is consistent with the relative branching order and estimated ages of sc1 and sc6 inferred from the phylogenetic analysis of genomic sequences (Fig. 1). The fit between the temporal pattern of the outbreak and the evolutionary origins of isolates provides further support to the hypothesis of genomic diversification during the outbreak. In addition, the shift from sc1 to sc6 as the dominant contributing sub-cluster may be indicative of ongoing adaptation or increasing pathogenicity of the outbreak strain.

**Mutation spectrum and DNA repair defects.** Three atypically divergent isolates were recognized. The isolates CSID 3000516276 and CSID 3015183683 likely represent remnants of early diverged branches. In contrast, isolate CSID 3015183688 was placed at the end of a long branch (Fig. 1), suggesting an acceleration of its substitution rate. This isolate was determined to have a mutation in its *mutS* gene, leading to a hypermutator phenotype (see Supplementary Method 3.1).

Excluding the hypermutator, 247 nucleotide positions (out of 3,411,033 in the 3,408 concatenated cgMLST gene alignments;

0.0072%) were polymorphic among the outbreak isolates. Similar nucleotide variation was demonstrated using the assembly-free approach, which detected 290 SNPs (out of 3,571,924 sites; 0.0081%). We further identified one 2 bp deletion, one 4 bp deletion, one 7 bp insertion, and five 1 bp deletions. This estimated evolutionary rate ($5.98 \times 10^{-6}$ substitutions per site per year within core genes, and $6.35 \times 10^{-6}$ substitutions per site per year over the entire genome) is exceptionally high for a single-strain bacterial outbreak. We, therefore, analysed the mutational spectrum within the outbreak and compared it with the spectrum of the other *E. anophelis* sublineages, using the assembly-free approach. Strikingly, 253 out of 290 (87%) nucleotide substitutions along the branches of the outbreak tree were G/C->T/A transversions. This is a highly unusual pattern of mutation, and was significantly different from the mutational spectrum in the wider *E. anophelis* tree (11% G/C->T/A; Fig. 3). We noted that the mutational spectrum within the outbreak corresponds to mutations caused by the oxidative lesion 8-oxodeoxyguanosine (8-oxodG), suggesting either mutagenic growth conditions for the strain resulting from a high-oxidative stress environment, or impairment of the base excision repair pathway for 8-oxodG (the 'GO system'), which corrects

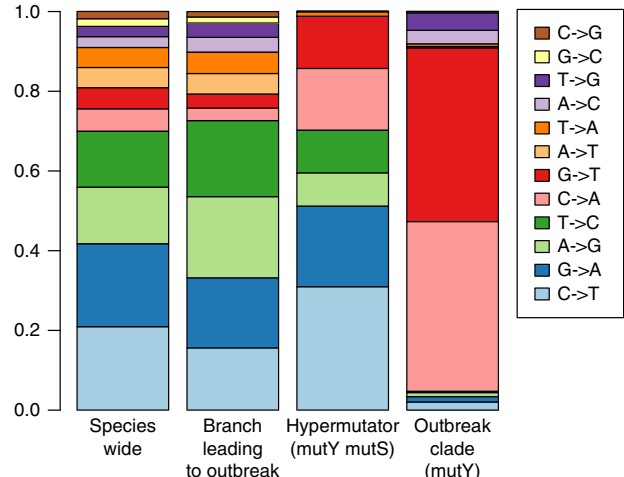

**Figure 3 | Mutation spectrum of *E. anophelis* strains by clade.** Frequency of each observed substitution mutation, reconstructed from FastML analysis, is shown for different parts of the *E. anophelis* tree.

these lesions[21–24]. We, therefore, inspected the genes that pertain to the GO system, and found that *mutY* was interrupted at position 841 in all outbreak isolates by the insertion of the 62,849 bp Integrative and Conjugative Element ICE*Ea*1 (for integrative and conjugative element 1 of *E. anophelis*, see below) (Fig. 4). This insertion resulted in a premature stop codon truncating the 57 terminal amino acids (aa) of the 342 aa-long MutY protein. MutY is an adenine glycosylase that functions in base excision repair to correct G-A mismatches[25]. Thus, MutY inactivation could explain the large number and atypical pattern of nucleotide substitutions observed within the outbreak. The ICE was not observed at this position in non-outbreak *E. anophelis* strains. Analysis of the mutational spectrum of substitutions on the branch leading to the outbreak strain (before its diversification started) revealed that it was very similar to that of the wider *E. anophelis* species tree (Fig. 3). This

indicates that the interruption of *mutY* via insertion of ICE*Ea*1 occurred shortly before the last common ancestor of the outbreak isolates.

ICE*Ea*1's integrase is 64% similar to the integrase of CTnDOT, a well-studied ICE[26,27]. We identified a potential integration site (TTT^TT) at position 841 of the *mutY* gene, flanked by inverted repeats in the ICE*Ea*1 and in the wild-type (WT) *mutY* gene (Fig. 4). We provide a model of the insertion of the ICE in a wild-type *mutY* gene (steps A and B, Fig. 4), which explains the position of the ICE in the outbreak strain. Simulating further steps of the ICE's lifecycle suggests that the ICE*Ea*1 insertion should be reversible and that the excision would reconstitute the original and functional *mutY* sequence (steps C and D, Fig. 4).

**Evidence for positive selection**. The atypical mutation spectrum attributed to the *mutY* truncation resulted in a very high non-synonymous to synonymous substitution ratio (ns/s = 21.4, excluding SNPs present only in the MutS- isolate), with most mutations causing amino-acid sequence alterations in the encoded proteins. Of the 49 nonsense mutations found in *mutS* competent isolates, 45 resulted from transversions unrepaired by the defective *mutY* (for example, GAA- > TAA, GAG- > TAG, and so on), including the *mutS* gene mutation resulting in the hypermutator phenotype of isolate CSID 3015183688. The substitution ratio of SNPs unique to this isolate (ns/s = 3.75) and its overall mutation spectrum (Fig. 3) were different from those of other outbreak isolates, as would be expected due to the high rate of base transition mutations in *mutS*-deficient isolates[28].

Among the 213 inferred protein changes (Supplementary Data 2, non-synonymous and nonsense mutations), some may have had important consequences regarding the virulence or resistance of the outbreak isolates, or on the fitness of the outbreak strain in its reservoir or source. We noted that the serine-83 of DNA gyrase *gyrA*, which is associated with quinolone resistance, was altered in one isolate (CSID 3000521792). Protein changes in the branch leading to node I, from which most outbreak isolates derived, may have contributed to the early adaptation of the outbreak strain to its reservoir or source. They occurred in genes

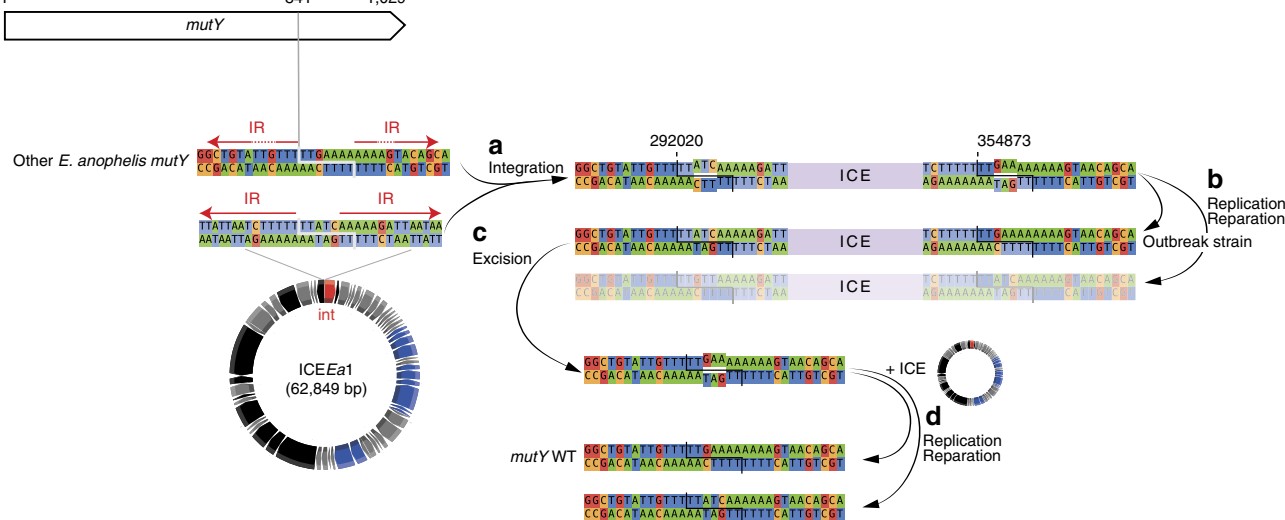

**Figure 4 | Excision of ICE*Ea*1 can lead to *mutY* WT in outbreak strains.** Here the insertion site is TTT^TT. In both the ICE and the *mutY*, there are inverted repeats (IR, red arrows) separated by ~5–6 nucleotides. Note that the chromosomal IRs are only partially conserved, as denoted by the interrupted arrows. (**a**) Upon insertion of the ICE at that site, this will create two heteroduplexes. (**b**) These will be resolved either by replication or by reparation. One of the two solutions to the heteroduplex resolution leads to the observed outbreak strain sequence. (**c**) If the ICE excises from the outbreak strain sequence, it will produce one heteroduplex. (**d**) The resolution of the heteroduplex left after excision of the ICE will lead to the *mutY* wild-type (WT) gene in one of the two scenarii.

coding for a TonB-dependent siderophore, a peptidase, a two-component regulator, a cysteine synthase and two ABC-transporters (Supplementary Data 2).

To detect positive selection during the course of the outbreak, we looked for genes with multiple parallel mutations. We found 27 genes that had two or more protein-altering mutations (either a non-synonymous or a nonsense mutation leading to protein truncation) that arose independently in separate branches of the tree. Prominent among these were three genes that each had five or six protein parallel mutations (Supplementary Data 2): the *wza* (A2T74_09840) and *wzc* (A2T74_09845) capsular export genes, and the gene A2T74_10040, which codes for a member of the SusD (Starch Utilization System) family of outer membrane proteins involved in binding and utilization of starch and other polysaccharides[29,30]. These observations are best explained by a strong selective pressure to abolish the function of the corresponding gene products. In light of the predominance of sub-cluster 6 towards the end of outbreak, it is interesting to note that the two changes that were specific to this sub-cluster (present in all 26 members of sc6, but in no member of other sub-clusters) were nonsense mutations in the genes *wza* and *susD* (Supplementary Data 2).

**Genomic features of the outbreak strain**. To define the unique genomic features of the outbreak strain, an analysis of the entire complement of protein families in *E. anophelis* genomes (that is, the *E. anophelis* pan-genome) was conducted (Supplementary Table 1). The *E. anophelis* pan-genome comprised 8,808 protein families, whereas only 3,637 protein families were observed among the 69 outbreak isolates (Supplementary Fig. 5). Further, the core-genome of the outbreak isolates represented 97% of the average number of proteins per genome, and 94% of the outbreak pan-genome. These results underline the strong homogeneity of the gene content of the outbreak isolates as compared with the extensive diversity observed within the *E. anophelis* species as a whole. Four isolates had a 77 kbp deletion affecting 75 genes; these were all from the same patient (Fig. 5; Supplementary Fig. 6; Table 1; Supplementary Data 3).

*E. anophelis* genomes are well known to harbour multiple genes putatively implicated in antimicrobial resistance. We found (Supplementary Data 4) that the outbreak isolates harboured resistance-associated genes previously observed in other *E. anophelis*[2,4,6,17], coding for multiple efflux systems, class A beta-lactamases, metallo-beta-lactamases and chloramphenicol acetyltransferase. Therefore, the Wisconsin outbreak strain possesses an array of antimicrobial genes similar to other *E. anophelis* strains, consistent with its multiple antimicrobial resistance phenotype (see below).

A search for putative virulence genes led to the identification of 67 genes (Supplementary Data 5). Among these, genes that were highly associated to the outbreak strain as compared with other *E. anophelis* isolates, included a CobQ/CobB/MinD/ParA nucleotide-binding domain protein located on the ICE*Ea*1 element (see below) and five genes involved in capsular polysaccharide synthesis. Capsules are important virulence factors of bacterial pathogens[31]. We, therefore, extended the search for other capsular synthesis associated genes (see Methods) and identified an identical Wzy-dependent capsular polysaccharide synthesis (*cps*) cluster in all outbreak isolates (Supplementary Fig. 7). As previously reported[2], the region of the *cps* locus that encodes for secretory proteins such as Wza and Wzc is highly conserved in *Elizabethkingia*, whereas the proteins involved in generating the specific polysaccharidic composition of the capsule are encoded in a highly variable

region (outbreak-specific region 5; Fig. 5). Within the 114 *Elizabethkingia* genomes, 17 different *cps* cluster types were defined based on their gene composition pattern (Supplementary Fig. 7). Remarkably, the Wisconsin strain shared its *cps* cluster (type I) with sublineage 2 isolates, which were associated with an earlier outbreak in Singapore[2,6]. This result suggests that horizontal gene transfer of the *cps* region between *E. anophelis* sublineages may drive the emergence of virulent lineages. The *cps* gene cluster type I has so far only been observed in these two human outbreak *E. anophelis* strains (that is, the Singapore outbreak[2,6] and the Wisconsin outbreak reported here). Altogether with our observation of multiple changes of the *cps* region during the diversification of the Wisconsin outbreak strain these data suggest a possible pathogenic role for the capsular polysaccharide in the outbreak strain.

To identify genomic regions unique to, or strongly associated with, the outbreak strain, we analysed the distribution of the pan-genome protein families within *E. anophelis* and found 13 gene clusters that were conserved among outbreak isolates (present in at least 67/69 outbreak isolates) but absent in most other *E. anophelis* sublineages (Fig. 5, Supplementary Fig. 6). The functional annotations of genes located in these genomic regions suggest they may confer to the outbreak strain improved capacities to tolerate heavy metals, acquire iron, catabolize sugars or urate and synthesize bacteriocins (Table 1; Supplementary Data 3).

Most notably, the integrative and conjugative element ICE*Ea*1 was present in all outbreak isolates but was absent in most other *E. anophelis* strains (region 2 in Fig. 5 and Supplementary Fig. 6). ICE*Ea*1 belongs to the *Bacteroidetes* type 4 secretion system (T4SS-B) class[32]. It encodes the full set of components required for integration/excision and conjugation, including an integrase (tyrosine recombinase), 12 genes coding for the type IV secretion apparatus (including a VirB4 homologue and the type IV coupling protein), a relaxase (MOBP1), an ATPase (virB4) and two genes encoding for RteC, the tetracycline regulator of excision protein (Supplementary Fig. 8). Among its cargo genes (Supplementary Data 3), ICE*Ea*1 carried genes putatively coding for a RND-family cation export system composed of a cobalt-zinc-cadmium efflux pump of the *czcA/cusA* family (which was affected by two distinct non-synonymous mutations during the outbreak), followed by genes with the following annotations: nickel and cobalt (*cnrB*) and mercury (*merC*) resistance, a P-type ATPase associated with copper export (*copA*), a receptor-binding hemin, a siderophore that may allow the bacteria to fix iron from the environment (*hemR*) and a solitary N-6 DNA methylase (MTase) that might be involved in protection from restriction systems. These annotations warrant future research on a possible contribution of the ICE*Ea*1 element to detoxification of divalent cations and to acquire iron from the host during infection. Within the wider *E. anophelis* genome set, the ICE*Ea*1 element was observed in only six non-Wisconsin outbreak isolates: four isolates associated with the Singapore outbreak and strains CIP 60.59 and NCTC 10588 (Supplementary Figs 8 and 9), which were both isolated from patients with severe human infections during the 1950's (Supplementary Data 1). The association of ICE*Ea*1 with virulence deserves further functional investigation. In the six other strains, the ICE*Ea*1 element was inserted in genomic locations distant from *mutY* (Supplementary Fig. 8b). We could not find any other mobile genetic element (that is, prophages, integrons and plasmids) in the genomes of outbreak strains.

Finally, one of the outbreak-associated genomic regions comprises genes for a sodium/sugar co-transporter, a xylose isomerase and a xylose kinase (region 9, Fig. 5, Supplementary Data 3). This region was also present in the mosquito gut isolates Ag1 and R26 (region 9, Supplementary Fig. 6, Supplementary Fig. 9)[11].

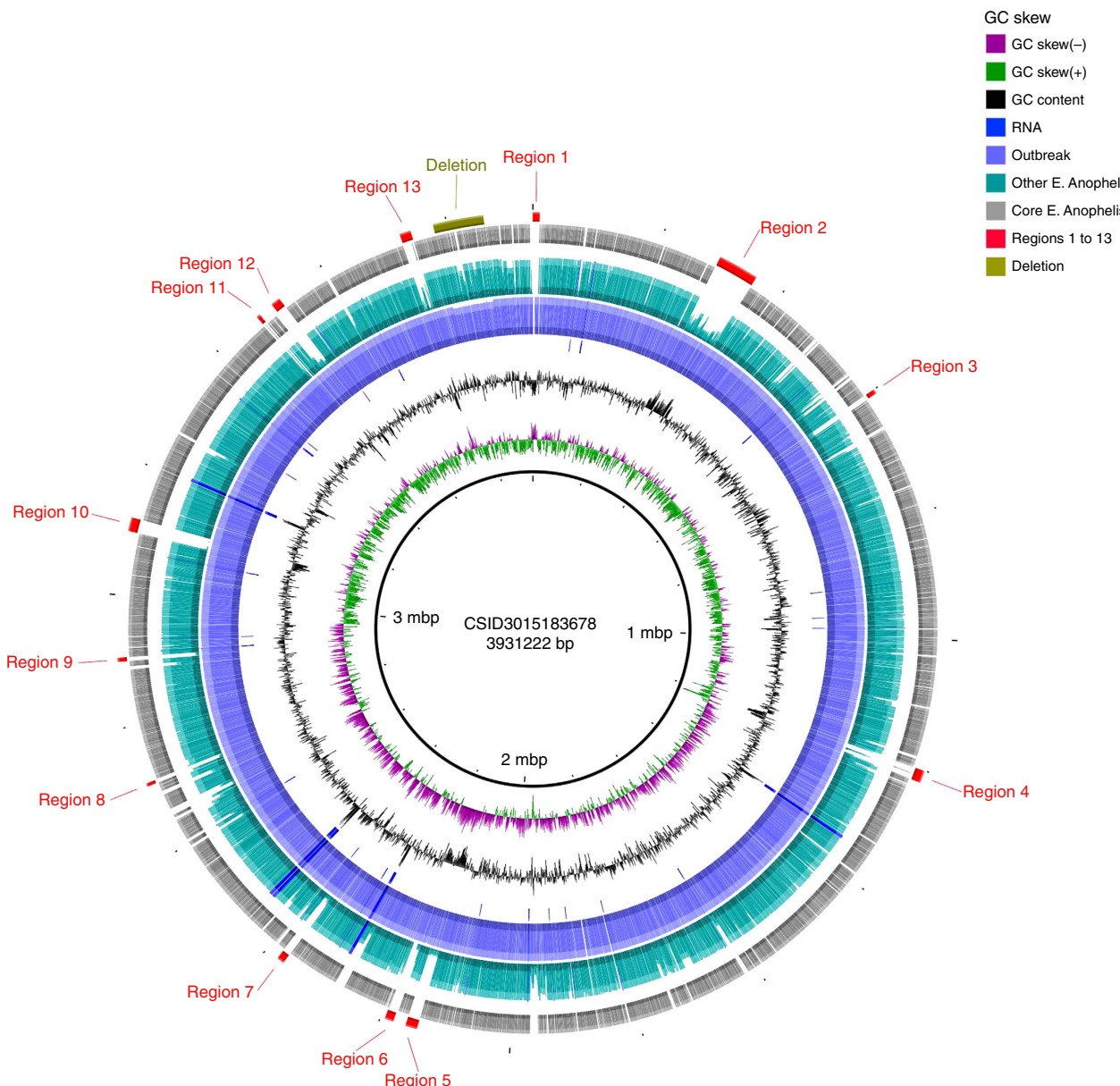

**Figure 5 | Circular representation of gene content variation between the outbreak strain and 30 other *E. anophelis* genomes.** Circles, from 1 (innermost circle) to 8 (outermost circle), correspond to: Circle 1: scale of the reference genome CSID 3015183678. Circle 2: GC skew (positive GC skew, green; negative GC skew, violet). Circle 3: G + C content (above average, external peaks; below average, internal peaks). Circle 4: non-coding genes (rRNA, tRNA, tmRNA); their positions are also reported in circles 5 and 6. Circle 5: frequency of CSID 3015183678 protein-encoding DNA sequences (CDSs) among the 69 outbreak isolates genomes; note the high conservation, except for a 77 kbp deletion near position 3.8 Mbp. Circle 6: frequency of CSID 3015183678 genes in all other *E. anophelis* genomes, revealing genomic regions containing CDSs with low frequency in the species as a whole. Circle 7: core genes in all 99 *E. anophelis* strains. Circle 8: remarkable genomic regions of the outbreak isolates; specific regions are marked in red, deletion in olive. Functional information about CDSs comprised in these regions is given in Table 1. The figure was obtained using BLAST Ring Image Generator (BRIG)[73]. For more details, see Supplementary Fig. 8.

**Antimicrobial susceptibility of outbreak isolates.** Antimicrobial susceptibility testing (Supplementary Data 6) revealed a strong homogeneity among outbreak isolates. A low susceptibility against most beta-lactams was found; isolates were resistant against ceftazidime and imipenem, but susceptible to piperacillin, piperacillin-tazobactam and cefepime. Outbreak isolates were also resistant to aminoglycosides (amikacin, gentamycin, tobramycin) and showed low *in-vitro* susceptibility to chloramphenicol, fosfomycin, tetracycline and vancomycin. These phenotypes demonstrate the high level of antimicrobial resistance of *E. anophelis* Wisconsin outbreak isolates, consistent with previous data

on other *E. anophelis* isolates[6,14,15,33]. In contrast, outbreak isolates were susceptible to quinolones (ciprofloxacin, levofloxacin) and showed high *in-vitro* susceptibility to trimethoprim-sulfamethoxazole and to rifampicin. Variation in resistance among outbreak isolates was found only for chloramphenicol and for quinolones: first, isolate CSID 3000521792 was resistant to quinolones, consistent with its amino-acid alteration at position 83 of DNA gyrase subunit A (Supplementary Data 2). Second, resistance of isolate CSID 3000516072 to chloramphenicol was decreased compared with other isolates (Supplementary Data 6). Interestingly, CSID

**Table 1 | Genomic features associated with the Wisconsin outbreak isolates\*.**

| Name | Start | End | Size (nt) | Size (CDS) | Remarkable features of genomic region |
|---|---|---|---|---|---|
| Region 1 | 3,926,747 | 10,253 | 10,564 | 11 | Type I restriction/modification system; DNA-invertase |
| Region 2 | 292,287 | 354,501 | 62,215 | 62 | ICEEa1; metal resistance, hemin receptor precursor; mercury resistance; enterobactin exporter |
| Region 3 | 599,595 | 606,529 | 6,935 | 5 | Tetratricopeptide repeat protein |
| Region 4 | 1,200,465 | 1,219,016 | 18,552 | 13 | CTP pyrophosphohydrolase |
| Region 5 | 214,2546 | 216,0415 | 17,870 | 17 | Putative polysaccharide synthesis clusters (capsule and LPS) |
| Region 6 | 217,9815 | 219,3156 | 13,342 | 13 | Putative polysaccharide synthesis clusters (capsule and LPS) |
| Region 7 | 2,367,659 | 2,378,760 | 11,102 | 8 | Putative deoxyribonuclease RhsC |
| Region 8 | 2,705,573 | 2,710,635 | 5,063 | 5 | Glycosyl hydrolase, beta-glycosidase and beta-glucosidase |
| Region 9 | 2,898,750 | 2,904,987 | 6,238 | 5 | Xylulose kinase, xylose isomerase, sodium/glucose co-transporter |
| Region 10 | 3,097,180 | 3,118,179 | 21,000 | 21 | Transposase; FAD-dependent urate hydroxylase (flavoprotein involved in urate degradation to allantoin) |
| Region 11 | 3,477,609 | 3,483,251 | 5,643 | 7 | Hypothetical proteins |
| Region 12 | 3,506,671 | 3,521,185 | 14,515 | 10 | Starch-binding outer membrane protein; Ferrienterobactin receptor precursor; Susd/RagB outer membrane lipoprotein; Nisin biosynthesis protein NisC; Putative lantibiotic biosynthesis protein |
| Region 13 | 3,727,334 | 3,744,334 | 17,001 | 15 | Transposase, IS200-like |
| Deletion† | 3,779,205 | 3,856,342 | 77,138 | 75 | Multidrug resistance protein MdtE and efflux pump membrane transporter BepE; HopJ type III effector protein (found in plant pathogens); ABC-2 family transporter protein; Cytochrome c551 peroxidase precursor; H(+)/Cl(−) exchange transporter ClcA; Sulfite exporter TauE/SafE; Bicarbonate transporter BicA; Vitamin B12 transporter BtuB precursor; Putative transporter YycB; beta-lactamase |

CTP, cytidine triphosphate; FAD, flavin adenine dinucleotide; LPS, lipopolysaccharide.
Positions refer to the genome of reference strain CSID 3015183678.
\*Present in at least 90% of outbreak genomes and in <20% of the other *E. anophelis*.
†Absent in four *E. anophelis* outbreak genomes (patient 30).

3000516072 had an arginine to leucine alteration at position 164 of the chloramphenicol acetyltransferase, which may impact the function of this chloramphenicol resistance enzyme. As compared with the African isolates, Wisconsin outbreak isolates were more resistant to cefoxitin, amikacin and isepamycin, but less resistant to chloramphenicol, rifampicin and tetracycline. Outbreak isolates differed from the Singapore isolates by their lower resistance level to macrolides and to isepamycin. However, in the absence of interpretive breakpoints for *Elizabethkingia anophelis* antimicrobial resistance, the clinical significance of the above differences is unclear.

## Discussion

We defined the phylogenetic diversity and genomic features of a strain of *E. anophelis* that caused an exceptionally large and primarily community-associated outbreak. Our phylogenetic analyses clearly established that the outbreak was caused by a single strain. The phylogenetic analysis showed that the outbreak strain represents a previously undescribed sublineage within *E. anophelis*. The nucleotide distance that separates the outbreak strain from the closest sublineages of *E. anophelis* with available genome data is nearly 1%, similar to the distance that separates, for example, clonal groups of *Klebsiella pneumoniae* with very distinct virulence properties[34,35]. These results raise the possibility that the sublineage to which the outbreak strain belongs may have evolved distinctive virulence or ecological properties, which could have contributed to the atypical size and community occurrence of the Wisconsin outbreak. For example, as xylose is one of the most abundant sugars on Earth, the genes for xylose utilization might provide a growth advantage to the outbreak strain in a reservoir, possibly in the presence of vegetation-derived nutrients. Although it is tempting to speculate on the possible link between the genomic features of the outbreak strain and the magnitude and setting of the outbreak, it is difficult to assess whether the strain has enhanced virulence in humans. The morbidity and mortality potentially attributable to *E. anophelis* infection was confounded by serious co-morbid conditions existing in patients affected by this outbreak. This work nevertheless suggests multiple avenues of research regarding the potential impact of the outbreak strain's unique capsule structure, cation detoxification capacity and sugar metabolism on its pathogenicity.

The phylogenetic position of *Elizabethkingia* strains selected for comparative purposes revealed the need for taxonomic reassignment for a large number of strains, as expected given recent taxonomic changes and the difficulty in differentiating *Elizabethkingia* species based on phenotypic characteristics. We found that several strains initially identified as *E. meningoseptica* are in fact *E. anophelis*. *E. anophelis* can be identified using matrix-assisted laser desorption ionization - time of flight (MALDI-TOF) analysis, but requires updated reference spectrum databases as found here and in a previous work[15]. This further indicates that the clinical importance of *E. anophelis* was previously underestimated, in agreement with results of a recent study[15]. These observations call for more research regarding *E. anophelis* ecology, epidemiology and virulence mechanisms.

Our results highlight important temporal and spatial patterns of the outbreak. They suggest that the bacteria may have been growing in a contaminated reservoir for nearly one year before the first infections occurred. No confirmed *E. anophelis* case could be retrospectively associated with the outbreak before November 2015. This suggests occurrence of either silent propagation resulting in human cases that remained undiagnosed or diversification of the strain in the unidentified source(s) before the initial infection of a patient. Further, the notable evolution of the pathogen during the outbreak, demonstrated by the temporal accumulation of substitutions, suggests that the source must be permissive to strain growth. Alternately, a long incubation period might precede the onset of disease, thus providing a possibility for the isolates to evolve within the patients, but the lack of diversity among multiple isolates from a single patient argues against this possibility. The uniformity of isolates from single patients also shows that although the outbreak strain has diversified, either patients were exposed to sources contaminated by a low-diversity

population, or the colonization and infectious process involves a bottleneck resulting in single clonal infection, even from a multi-contaminated source. This work thus provides a striking additional example of the now well-established power of genomic sequencing to facilitate critical re-examination of epidemiologic hypotheses and outbreak patterns[36,37].

Outbreak isolates differed by a large number of polymorphisms, and the spectrum of mutations among the outbreak isolates was unlike normal variation among other *E. anophelis*. Much less diversity is typically observed during bacterial outbreaks lasting less than one year[37,38]. Because the intra-outbreak diversity was so unusual, we confirmed it by two independent approaches: gene-by-gene analysis (cgMLST) and mapping-based SNP analysis. We identified a probable cause of this atypical mutation pattern: the disruption of the *mutY* gene coding for adenine glycosylase. Anecdotally, one strain further developed a hypermutator phenotype through a disruption of its *mutS* gene, which encodes a nucleotide-binding protein involved in the DNA mismatch repair system.

Beneficial mutations in the outbreak strain could have been selected under conditions encountered in the reservoir or the source, or during colonization or infection. Our results strongly suggest that disruptions of genes encoding proteins involved in polysaccharide utilization or capsule secretion were positively selected. Multiple outbreak isolates had alterations in the starch utilization SusD protein, and/or partial or complete disruption of either the Wza or Wzc polysaccharide transport proteins. The success of sub-cluster 6 during the later weeks of the outbreak might have resulted from the combined effect of complete disruption of both SusD and Wza. How the disruption of these functions could result in a competitive advantage for the outbreak isolates is not immediately apparent. We can speculate that the loss of capsular polysaccharides may facilitate adhesion and colonization, lead to reduced antigenicity or allow the bacteria to disperse more readily due to modified adherence to surfaces. Regardless, our results depict a dynamic outbreak strain that continued evolving while the outbreak was ongoing. One notable outcome of the exceptional genome dynamics of the outbreak strain was the replacement of sub-cluster 1 by sub-cluster 6 as the dominant subtype infecting the patients.

It is likely that the *mutY* phenotype resulted in an increased adaptive capacity of the outbreak strain. For example, the short-term advantage conferred by mutator phenotypes was previously documented in *Pseudomonas aeruginosa* infections among patients with cystic fibrosis[39]. Therefore, the integration of the ICE*Ea*1 in the *mutY* gene was likely favoured by hitchhiking with a positively selected mutation caused by the lack of this repair mechanism. In the longer run, defective DNA repair genes are counter-selected because of mutational load or because they diverge from optimal fitness peaks once the environment is stabilized[40,41]. Based on the structure of the integration site, we hypothesize that the outbreak strain could revert to a functional *mutY* sequence by losing the ICE*Ea*1 through excision, thus recovering a full capacity to repair DNA. This reversible switch of hyper-mutagenesis might have important implications regarding the future survival and possible resurgence of the Wisconsin outbreak strain. We, therefore, urge healthcare and public health systems to establish a laboratory based surveillance for *Elizabethkingia* infections, and to be particularly vigilant for a possible re-emergence of the unique *E. anophelis* strain that caused the Wisconsin outbreak.

## Methods

**Bacterial isolates.** Wisconsin clinical laboratories were asked to submit any confirmed or suspect *Elizabethkingia* isolates to Wisconsin State Laboratory of Hygiene for identification and pulsed-field gel electrophoresis subtyping. Isolates were initially identified as *E. meningoseptica* using conventional biochemical assays and the Bruker MALDI-TOF spectral library. Pulsed-field gel electrophoresis subtyping using an in-house developed protocol, modified after consultation with CDC, was used to determine genetic relatedness among all suspect outbreak isolates. All isolates determined to be *Elizabethkingia* species were submitted to CDC for further characterization. Upon arrival, bacteria were cultivated on heart infusion agar supplemented with 5% rabbit blood agar at 35 °C. The outbreak strain isolates were correctly identified as *E. anophelis* using an expanded MALDI-TOF spectral library, genome sequencing and optical mapping. Conventional biochemical testing was restricted to oxidase, catalase and Gram stain after the MALDI-TOF spectral library provided by the CDC Special Bacteriology Reference Laboratory proved to be a reliable method of identification.

Outbreak isolates (Supplementary Data 1; labelled as Wisconsin outbreak) were primarily derived from blood (54 isolates), and also from sputum (3), bronchial wash (3), pleural fluid (1), synovial fluid (1) and other sites (7) from patients residing in 12 different counties in Southeast Wisconsin, 1 county in Illinois and 1 county in Michigan. Specimen collection dates ranged from November 2015 through March 2016. DNA was extracted using the Zymo Fungal/Bacterial DNA Microprep Kit (Zymo Research Corporation, Irvine, CA). Libraries were prepared using the NEBnext Ultra DNA Library Prep Kit for Illumina (New England Biolabs, Ipswich, MA), and sequence reads were generated with the Illumina MiSeq Reagent Kit v2 and MiSeq instrument (Illumina, Inc., San Diego, CA).

For comparative purposes, we included seven isolates stored in the Pasteur Institute's collection (Collection de l'Institut Pasteur or CIP; Supplementary Data 1, isolate names starting with CIP). Strains were cultivated on trypticase soy agar at 30 °C. DNA extraction was performed using the MagNA Pure 96 robotic System with the MagNA Pure 96 DNA and Viral Nucleic Acid small volume kit (Roche Diagnostics). Libraries were constructed using the Nextera XT DNA Library Preparation kit (Illumina, Inc., San Diego, CA) and sequenced on a NextSeq-500 instrument using a $2 \times 150$ paired-end protocol.

We also downloaded and included all *Elizabethkingia* genome sequences ($n = 28$ as of 28th April 2016) and sequencing read data sets ($n = 10$ as of 28th April 2016) available in sequence repositories (Supplementary Data 1).

The complete 114 *Elizabethkingia* isolate data set contained 69 Wisconsin outbreak isolates from 59 different patients (one to four isolates per patient, see Supplementary Data 1), 29 historical *E. anophelis* strains, one strain initially classified as *E. endophytica* that has been shown to belong in fact to *E. anophelis*[20], 5 *E. meningoseptica* strains, and 10 strains that belonged to the *E. miricola* cluster (see Results and Supplementary Fig. 1).

**Genome assembly and annotation.** For each outbreak isolate, an initial assembly was generated using the Celera De Bruijn graph assembler (Celera Genomics Workbench v8, Alameda, California). Isolate CSID 3015183678 was selected as reference for comparative genomics analyses because of its central position in an optical mapping cluster analysis of early outbreak isolates. Its contigs were ordered and oriented based on the *Nco*I optical map to generate a complete circularized genome, which was confirmed based on a PacBio genome sequence[42]. Complete circularized genomes from the other outbreak strain isolates were generated by mapping reads to the reference genome using CLC Genomics Workbench v8 (CLC bio, Waltham, MA), and manually aligned using BioEdit[43]. Indels in the circularized genomes were located using BioEdit's Positional Nucleotide Numerical Summary function.

Assemblies of the seven genomes from the CIP and from publicly available data sets for which only sequence reads were available (see Supplementary Data 1), were generated using SPAdes v.3.6.2 (ref. 44) on pre-processed reads, that is, trimming and clipping with AlienTrimmer v.0.4.0 (ref. 45), sequencing error correction with Musket v.1.1 (ref. 46), and coverage homogenization with khmer v.1.3 (ref. 47).

To obtain uniform and consistent annotations for core and pan-genome analyses, all 114 genome sequences were annotated using PROKKA v.1.11 (ref. 48). The main characteristics for each genome assembly are described in Supplementary Data 1. However, in discussion of the various loci throughout this paper, the locus tags from NCBI's Prokaryotic Genome Annotation Pipeline annotation of reference isolate CSID 3015183678 are used.

**Core-genome identification.** We built two core-genomes (that is, sets of orthologous proteins present in all genomes compared). The first one contained the proteins common to all *E. anophelis* genomes, while the second one contained the proteins common to all outbreak genomes. Orthologues were identified as bidirectional best hits, using end-gap free global alignment, between the reference outbreak proteome (CSID 3015183678) and each of the 98 other *E. anophelis* proteomes (for the *E. anophelis* core-genome) or each of the 68 other outbreak proteomes (for the outbreak core-genome). Hits with less than 80% similarity in amino-acid sequence or $>20\%$ difference in protein length were discarded. Because genomes from the same species typically show low levels of genome rearrangements at these short evolutionary distances, and horizontal gene transfer is frequent, proteins outside a conserved neighbourhood shared by different strains are likely to be xenologs or paralogues. Thus, for each of the previous pairwise comparisons, the list of orthologues was refined using information on the conservation of gene neighbourhood. Positional orthologues were defined as bidirectional best hits adjacent to at least four other pairs of bidirectional best hits within a

neighbourhood of ten genes (five upstream and five downstream). Finally, only the proteins having positional orthologues in 100% of the compared genomes (all *E. anophelis* genomes or all outbreak genomes) were kept. This resulted in a total of 2,530 proteins for the *E. anophelis* species core-genome, and 3,434 proteins for the Wisconsin outbreak core-genome (see Supplementary Fig. 5).

**cgMLST analysis.** For the core-genome MLST (cgMLST) analysis, we used two cgMLST schemes (sets of genes present in most isolates and selected for genotyping): one for the *Elizabethkingia* genus, and one for the Wisconsin outbreak isolates. The *Elizabethkingia* cgMLST scheme used was reported previously[2] and contains 1,546 genes. For the novel, Wisconsin outbreak cgMLST scheme, we started from the list of positional orthologues of the outbreak genomes (described above, in the outbreak core-genome part), and added the following conditions to ensure maximum discriminatory potential for genotyping purposes. First, we used the protein-coding sequences (coding DNA sequence, or CDS) having positional orthologues in at least 80% of the outbreak genomes. The use of this lower threshold (instead of 100% for the core-genome), allowed the use of more markers. Next, we removed from this list very small CDS (<200 bp) and genes with closely related paralogues (genes in the same genome with >80% similarity in amino-acid sequence and <20% difference in protein length). All genes already present in the *Elizabethkingia* cgMLST scheme were also discarded. These resulted in a set of 1,862 genes for the Wisconsin cgMLST scheme. These protein-coding genes, together with the 1,546 genes of the genus cgMLST scheme, constitute a total of 3,408 loci used for genotyping Wisconsin outbreak isolates of *E. anophelis*. The two cgMLST schemes are implemented in the Institut Pasteur instance of the BIGSdb database tool[49]. Allele sequences and their corresponding numerical designations are publicly accessible at http://bigsdb.pasteur.fr.

**Phylogenetic analysis of cgMLST data.** CDSs corresponding to the cgMLST schemes loci were aligned at the amino-acid level with MAFFT v.7.245 (ref. 50), back-translated to obtain multiple codon-based sequence alignments, and finally concatenated to obtain supermatrices of characters. This procedure was performed for (i) the entire *Elizabethkingia* sample (114 genomes) with the genus cgMLST scheme (1,546 loci, 554,224 aligned codons), and (ii) the Wisconsin outbreak isolates (69 genomes) by adding the dedicated cgMLST scheme to the genus one (total of 3,408 loci, 1,137,011 aligned codons). For each supermatrix of characters, the phylogenetic analysis was performed using IQ-TREE[51] with the codon evolutionary model being selected to minimize the BIC criterion, that is, $GY+F+\Gamma_4$ and $GY+F$[52] for the *Elizabethkingia* and for the Wisconsin outbreak samples, respectively.

**Mapping-based SNP analysis.** To assess variation of the entire genome including intergenic regions for phylogenetic analysis of the outbreak isolates, we used a read mapping approach. All read sets were mapped against the same reference outbreak genome sequence (CSID 3015183678) as used for core-genome and cgMLST locus definitions. Read mapping, SNP calling and preliminary filtering were completed using the RedDog phylogenomics pipeline (https://github.com/katholt/RedDog)[53]. Because we were primarily interested in phylogenetic analysis of the conserved regions of the *Elizabethkingia* genomes, SNP sites at which mapping and base calling could be confidently conducted in <95% of isolates were excluded from further analysis (most of these were located in a 77 kbp region that was deleted in four isolates that were derived from the same patient), as were SNPs located in either putative phage-associated or repeated regions of the reference genome, as detected by Phaster[54] or the nucmer algorithm of MUMmer v3 (ref. 55), respectively. We initially identified 467 SNP loci among the 69 outbreak isolates, and generated an alignment of concatenated SNP alleles at these loci. The spatial distribution of SNPs was visually inspected using Gingr[56]. A ~2 kbp cluster of SNPs was identified (density >0.1, compared with density <0.01 across the rest of the genome), affecting the protease A2T74_14135 in a subset of isolates. Spatially clustered SNPs are typically introduced together via homologous recombination and thus reflect horizontal rather than vertical evolution; hence, this region was excluded from phylogenetic analysis. This yielded a final set of 374 SNPs representing changes that arose within the population of outbreak isolates, within a total core-genome of 3,571,924 bp in size (90.9% of the reference sequence).

The concatenated alignment of these SNP alleles was used to generate a maximum likelihood phylogenetic tree for the outbreak isolates using IQ-TREE[51] (see Supplementary Fig. 2, Supplementary Method 3.2 and Supplementary Data 7). SNPs were mapped back to the tree using FastML v3.1 and the details of each substitution mutation (branch, ancestral allele, derived allele) were extracted from the marginal sequences output file (Supplementary Data 2). The coding effects of the SNPs, inferred using the annotated reference genome, was defined using the parseSNPtable.py script in RedDog and analysed using R.

**BEAST analyses.** Date estimates of all nodes were derived using BEAST v.2.3.1 (refs 57,58) on the cgMLST supermatrix of aligned nucleotide characters (Supplementary Data 8) with the $GTR+\Gamma_4+I$ nucleotide evolutionary model (one per codon position) and lognormal relaxed-clock model. Constant population size was selected as a tree prior, and BEAST was run with $10^8$ chains in order to obtain large effective sampling size values. For comparison, the BEAST analysis was

also conducted on the SNP alignment (Supplementary Data 9), using a HKY substitution model and a lognormal relaxed-clock model with constant population size (Supplementary Method 3.3). The significance of the temporal signal in each analysis was assessed using the tip-date randomization technique[59–61] based on 30 samples with reshuffled dates.

**Pan-genome analysis.** Pan-genomes were built by clustering homologous CDSs into families. We determined the lists of putative homologs between pairs of genomes with BLASTP v.2.0 and used the E-values (<10$^{-4}$) to perform single-linkage clustering with SiLiX v.1.2 (ref. 62). A CDS was included in a family if it was homologous to at least one CDS already in the family. SiLiX parameters were set to consider two CDSs as homologs if their aligned part had at least 60% (*Elizabethkingia* genus) or 80% (*E. anophelis*) sequence identity and included >80% of the smallest CDS. The pan-genomes of *Elizabethkingia* and of the outbreak isolates were determined independently.

**Detection of capsular gene clusters.** To identify capsular gene clusters, we used our previous approach[2]. In brief, we performed a keyword search of the Pfam database v.29.0 (http://pfam.xfam.org) for protein profiles involved in capsular polysaccharide production such as glycosyl transferases, ABC transporters, Wzx flippase and Wzy polymerase. We then performed a search of these profiles in *Elizabethkingia* genomes in HMMER3 v.3.1b1 (ref. 63) with the E-values <10$^{-4}$ and a coverage threshold of 50% of the protein. After the identification of a putative capsular cluster across all genomes, several proteins within the cluster did not match any of the previously selected protein profiles. For completeness, we searched these proteins for known functional domains against the PFAM database using the command hmmscan included in the software HMMER3, and recorded their family and/or annotation (see Supplementary Fig. 7, and regions 5 and 6 of Supplementary Data 3).

**Antimicrobial resistance and virulence-associated genes.** Acquired antimicrobial resistance genes were detected using HMMER3 v.3.1b1 to screen genome sequences against the ResFams (Core v.1.2), a curated database of antimicrobial resistance protein families and associated profile hidden Markov models with the cut_ga option[64] (Supplementary Data 4). Virulence-associated genes were identified by screening genome sequences against the VFDB 2016 (ref. 65) using BLASTP v.2.0 (minimum 40% identity with E-value <10$^{-5}$), as in (ref. 5) (Supplementary Data 5).

**Detection of mobile genetic elements.** ICEs were identified and classified using MacSyFinder v.1.0.2 (ref. 66) with TXSScan profiles[67]. CRISPR-Cas systems were searched using MacSyFinder v.1.0.2 with Cas-Finder profiles[66] and CRISPR-Finder[68], with default parameters. Integrons were searched using IntegronFinder v.1.4 with –local_max option[69], and prophages using VirSorter v.1.0.3 on RefSeqDB only[70] and PhageFinder v.4.6 (ref. 71).

**Antimicrobial susceptibility testing.** Antimicrobial susceptibility testing was performed by Kirby Bauer disk diffusion method (http://www.eucast.org/fileadmin/src/media/PDFs/EUCAST_files/Breakpoint_tables/v_6.0_Breakpoint_table.pdf)[72]. As no interpretative criteria exist for *Elizabethkingia,* results were interpreted according to European Committee on Antimicrobial Susceptibility Testing (EUCAST) criteria for *Pseudomonas spp.* We tested a broad range of antibiotics: beta-lactams (piperacillin, cefotaxime, ceftazidime, imipenem, ampicillin, amoxicillin, amoxicillin-clavulanic acid, cephalexin, cefuroxime, cefoxitin, cefepime, cefoperazone-sulbactam, piperacillin-tazobactam), aminoglycosides (streptomycin, amikacin, isepamycin, tobramycin, gentamicin, kanamycin), quinolones (nalidixic acid, ciprofloxacin, pefloxacin, levofloxacin, moxifloxacin), macrolides (erythromycin, clarithromycin, spiramycin, azithromycin) and other classes (chloramphenicol, sulfamethoxazole-trimethoprim, fosfomycin, rifampicin, linezolid, tetracyclin, vancomycin and tigecyclin).

**Data availability.** Reads for all outbreak isolates and complete genome sequences of outbreak isolates CSID 3015183678, CSID 3015183684, CSID 3000521207 and CSID 3015183681 were submitted to NCBI, associated with project ID PRJNA315668. Reads and draft genome sequences for strains Po0527107 and V0378064 (ref. 2) were submitted to the European Nucleotide Archive and are available under their respective project IDs, PRJEB5243 and PRJEB5242. Reads for strains CIP 78.9, CIP 60.59, CIP 104057, CIP 108654, CIP 79.29, CIP 80.33 and CIP 108653 were submitted to the European Nucleotide Archive and are available under project ID PRJEB14302. In addition, every genome sequence assembled during this study is available in the Institut Pasteur instance of the BIGSdb database tool dedicated to *Elizabethkingia* (http://bigsdb.web.pasteur.fr/elizabethkingia). Supplementary data, tables and high resolution figures are available through FigShare at this link (https://doi.org/10.6084/m9.figshare.c.3674146.v5). We also created a project on microreact, available at this link: https://microreact.org/project/SyaeGCjvg.

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

## Acknowledgements

We thank D. Mornico of Institut Pasteur and V. Nyak of CDC for assistance with submission of sequence data to public repositories. We would also like to thank the State Health Departments of Michigan and Illinois for contributing strains and information for the cases outside of the State of Wisconsin. The efforts of laboratory staff in both DHQP and DHCPP are greatly appreciated. This work was supported by Institut Pasteur, French government's Investissement d'Avenir program Laboratoire d'Excellence 'Integrative Biology of Emerging Infectious Diseases' (grant ANR-10-LABX-62-IBEID), and the Advanced Molecular Detection (AMD) initiative at CDC. O.R. was supported by a fellowship from Fondation pour la Recherche Médicale (grant number ARF20150934077). The findings and conclusions in this report are those of the authors and do not necessarily represent the official position of the Centers for Disease Control and Prevention.

## Author contributions

This project was designed by K.E.H., J.R.M. and S.B. Specimens and epidemiologic data were collected by K.M.G., T.M., D.W., L.I.E., M.S.W., M.B.C., J.N.-W., G.B. and J.P.D. Whole-genome sequences were generated and assembled by A.C.N., A.M.W., M.E.B., O.R., J.C., D.C., A.C. and V.E. Optical mapping was done by V.L. and P.J. cgMLST analysis was done by A.C. and E.L. Read mapping and SNP analysis was done by A.C.N., K.E.H. and D.J.E. Core and pan-genome analyses were done by A.P. and M.T. Capsular cluster analysis was done by O.R. Analysis of the ICE*Ea*1 integration site in *mutY* was performed by A.C.N. and J.C. Phylogenetic and BEAST analyses were done by D.J.E., K.E.H. and A.C. Antimicrobial susceptibility testing was performed by P.H. and S.B. Additional data analyses and figure creation were done by A.P., A.C.N., A.C., M.T., C.A.G., K.E.H. and S.B. Overall coordination of the study and of manuscript writing was done by S.B. The manuscript was drafted and edited by A.P., E.L., A.C.N., K.E.H., T.M., D.J.E., C.A.G., M.S.W., M.T., E.P.C.R., J.P.D., J.R.M. and S.B. All authors provided final approval of the version submitted for publication.

## Additional information

**Competing interests:** The authors declare no competing financial interests.

