## [Peer Review File · Nature Communications]

Reviewer #1 (Remarks to the Author):

This is an intriguing and highly anticipated report on the population genomic structure of a high-profile and quite mysterious *Elizabethkingia* outbreak in the United States. The authors have shown convincingly that the outbreak was caused by a single strain of *E. anophelis*. Surprisingly, the diversity within the outbreak and evolutionary rate was much greater than expected in a typical bacterial outbreak of this size. This high rate of change is convincingly attributed to a likely hypermutator phenotype as suggested by the disruption of a DNA repair gene.

The work is done to a technically very high standard and datasets have been made available. There are no major concerns.

My comments are all minor:

- It has been noted here and in other reports that a point source has not been identified for this outbreak. Could the authors expand in the discussion what the likely (or unlikely) sources might be, by reference to the phylogenetic data. The appearance of clonal strains in patients who were sampled multiple times, plus the high diversity across the outbreak perhaps points to sampling from a reservoir containing high levels of diversity (could it be water?).
- It would be interesting to contextualise this style of population structure with other outbreaks to look for similarities (*E. coli* O104:H4 is mentioned).
- In the abstract there is a reference to an elevated mutation rate — although quite likely, I cannot see that this has been experimentally determined.
- The reason for species misassignment within the *Elizabethkingia* genus is likely to relate to inadequate MALDI-TOF databases employed in microbiological diagnostic laboratories. This could be mentioned.

Reviewer #2 (Remarks to the Author):

This manuscript by Perrin et al. describes the genomic composition of *Elizabethkingia anophelis* isolates involved in a recent outbreak in Wisconsin. It represents a very robust assessment of these bacterial isolates and puts them in context with previously published sequences. This study identified high genetic diversity in these outbreak isolates, 6 subclusters within the outbreak as well as an ICEEa1 element integrated into the *mutY* gene, truncating it by 57aa (after 342aa). The authors postulate that this disruption a DNA repair gene might have contributed to the relatively high mutation rate. While this is an intriguing possibility this observation remains descriptive and lacks functional investigations to support this hypothesis. Alternatively, the manuscript can be strengthened by using the genomics presented here and apply these to shed further light on the putative sources of the evolving outbreak.

Major comments:

1. The first paragraph of the introduction lacks any citations. If these are unpublished data regarding the outbreak they need to be included as part of the results in the current paper.
2. ICEEa1 element: the text is somewhat inconsistent. Line 306 states that ICEEa1 was unique to the outbreak isolates. Line 334 notes that "...it was present in all outbreak isolates but was absent in most others". Line 354 states that it was "only present in 4 isolates of the Singapore outbreak and 2 others." This actually accounts for a sizable number of previously published sequences (6/30; 20% in Figure S11 – is that the correct denominator?) unrelated to the current outbreak. What was the insertion site in those other isolates (line 231 states it was not present at that position in other isolates).
3. One of the intriguing features associated with ICEEa1 is the potential nucleotide mutation bias.

However – what was the distribution of SNPs in the non-outbreak ICEEa1-positive isolates? The text does not clearly specify but it appears that these isolates did not exhibit this bias.

4. Paragraph starting at line 334: interesting observations but without experimental validation of any of the proposed processes, this remains highly speculative.

5. This study does not provide any additional clarity on the epidemiological sources of the outbreak. While the data suggest that there was one most common recent ancestor ~1 year prior to the first cases the manuscript demonstrates at least 6 different clusters. This genotypic information should now be used to inform the epidemiological investigations and focus on identifying sources of the clusters rather than the total outbreak.

Minor comments:

1. Line 362 – 364: speculative; should move into the discussion

2. Line 211 / 212: The German E. coli outbreak in 2011 was from a single source. There is no indication that this E. anophelis outbreak can be traced to direct exposure from a single source and as such the two cannot be directly compared.

3. Paragraph line 298: please state the major classes of anti-microbial resistance observed in the isolates.

Reviewer #3 (Remarks to the Author):

The authors present a comprehensive and clear analysis of the unique Elizabethkingia anophelis (EA) outbreak recently reported in US Midwest. 69 EA isolates from the outbreak were sequenced and analysed in the context of 45 other available Elizabethkingia genomes. The authors' major conclusions include: the outbreak was caused by a single EA sublineage that resolved into six unique subclusters, little to no within-host genetic diversity was observed within patients, the EA outbreak sublineage accrues ~24 SNVs per year and was estimated to have arisen ~1yr before the outbreak, and that this high mutational rate is likely attributable to a truncated MutY common to all outbreak isolates. A variety of genomic features of the outbreak strain are also reported, from virulence genes to conjugative elements.

This was a tremendously well-written paper describing a very carefully conducted series of analyses. The expertise of this reviewer lies largely in the genomic correlates of epidemiological trends and events, so my few minor suggestions/comments will attend to that area, rather than the comparative genomics work.

1. Figure 2 makes the spatio-temporal trends in the outbreak quite clear, but it would be nice to additionally link that back to Figure 1. Between Figure 1 and 2, there's no labelling to indicate which letter belongs to which country. Perhaps a more informative node labelling scheme for Figure 1 would be to assign a single colour to geography and use saturation to indicate an isolate's distance from the bulk of cases, which seem to be in Milwaukee. All Milwaukee-area strains would be, say, 100% blue, with the saturation decreasing with increasing geographic distance from Milwaukee. This would help give some spatial context to the tree in Figure 1.

2. The phylogenetic tree seems to suggest independent acquisition from some reservoir; however, there is an interesting pair in CSID3000521200 & CSID3000516360, whose genomes are identical. Is there any epidemiological information on these cases? Similarly, CSID3000521202 and CSID3000516028 are near-identical, and both come from the same pair of counties as the previously mentioned pair. Would be interesting to see some commentary on this, as well as any available information on which cases within a given county were associated with specific facilities.

3. The authors note that the outbreak EA sublineage contains a number of antimicrobial resistance determinants; a brief comment on how these relate to the course of antibiotics used in treating these patients would be interesting for the clinically-oriented reader.

4. Thank you for sharing the tree! It might also be helpful to include the BEAST XML file that was used to derive the time-labelled tree.

5. In a similar sharing vein, given the geographical and temporal nature of the data, it would be well-suited for presentation on microreact.org!

6. One last data-sharing comment - it might be helpful to mirror the supplemental tables at a site such as FigShare, rather than just on a single Dropbox folder. Because the supplemental tables are only available at the Dropbox folder, if that folder disappears, so does a very critical set of data from this analysis.

7. Line 237: what is CTnDOT?

Rebecca Furlong
Senior Editor
Nature Communications

Dear Editor,

Thank you very much for handling our manuscript. We hereby submit the revised version. Please find enclosed the point-by-point response to the reviewer's comments.

Reviewer #1 (Remarks to the Author):

This is an intriguing and highly anticipated report on the population genomic structure of a high-profile and quite mysterious *Elizabethkingia* outbreak in the United States. The authors have shown convincingly that the outbreak was caused by a single strain of *E. anophelis*. Surprisingly, the diversity within the outbreak and evolutionary rate was much greater than expected in a typical bacterial outbreak of this size. This high rate of change is convincingly attributed to a likely hypermutator phenotype as suggested by the disruption of a DNA repair gene.

The work is done to a technically very high standard and datasets have been made available. There are no major concerns.

Our answer: thank you very much for this positive appreciation.

My comments are all minor:

— It has been noted here and in other reports that a point source has not been identified for this outbreak. Could the authors expand in the discussion what the likely (or unlikely) sources might be, by reference to the phylogenetic data. The appearance of clonal strains in patients who were sampled multiple times, plus the high diversity across the outbreak perhaps points to sampling from a reservoir containing high levels of diversity (could it be water?).

Our answer: The extensive epidemiologic investigation of this outbreak was conducted by the Wisconsin Division of Public Health (WDPH) and the Centers for Disease Control and Prevention (CDC). This included testing a wide range of hypotheses. A report of the investigation and its results is included in a manuscript currently in preparation, primarily authored by the WDPH and CDC, and intended for a journal that reaches the public health and clinical communities. Therefore, we have written "publication in preparation" in the first paragraph of introduction. We also added two URLs and quote the abstract of a poster presented at IDweek 2016 congress (Eldabawi et al).

— It would be interesting to contextualise this style of population structure with other outbreaks to look for similarities (*E. coli* O104:H4 is mentioned).

Our answer: We are not clear regarding what is being requested. Nonetheless, while the information requested might be of interest, the reviewer considered this comment

to be minor and we feel this would add complexity to our manuscript and may better be the subject of another, interesting comparative publication of distinct outbreaks.

— In the abstract there is a reference to an elevated mutation rate — although quite likely, I cannot see that this has been experimentally determined.

Our answer: We have replaced ‘mutation rate’ by ‘evolutionary rate’ to make clear we refer to our calculated evolutionary rate.

— The reason for species misassignment within the *Elizabethkingia* genus is likely to relate to inadequate MALDI-TOF databases employed in microbiological diagnostic laboratories. This could be mentioned.

Our answer: The reviewer is right. To make this clear, we have added the following in the discussion section on misidentifications, “*E. anophelis* can be identified using MALDI-TOF analysis, but requires updated reference spectrum databases as found here and in a previous work (Lau 2016).”

Reviewer #2 (Remarks to the Author):

This manuscript by Perrin et al. describes the genomic composition of *Elizabethkingia anophelis* isolates involved in a recent outbreak in Wisconsin. It represents a very robust assessment of these bacterial isolates and puts them in context with previously published sequences. This study identified high genetic diversity in these outbreak isolates, 6 subclusters within the outbreak as well as an ICEEa1 element integrated into the *mutY* gene, truncating it by 57aa (after 342aa). The authors postulate that this disruption a DNA repair gene might have contributed to the relatively high mutation rate. While this is an intriguing possibility this observation remains descriptive and lacks functional investigations to support this hypothesis. Alternatively, the manuscript can be strengthened by using the genomics presented here and apply these to shed further light on the putative sources of the evolving outbreak.

Our answer: Thank you for your general appreciation. We recognize that the hypothesis of an implication of the *mutY* disruption in the high mutation rate is not proven. After evaluating the possibility of an experimental demonstration, we concluded it would represent a project in itself, with no guarantee of success, to try to excise the ICE or transfer it to a *mutY* wildtype strain. Our work is centred on the outbreak analysis and we believe we were prudent enough in formulating that this is only a hypothesis. We now provide further details on the location of the ICE in other *E. anophelis* strains that harbour it, further underlining the uniqueness of the outbreak strain in having a disrupted *mutY* gene.

Major comments:

1. The first paragraph of the introduction lacks any citations. If these are unpublished data regarding the outbreak they need to be included as part of the results in the current paper.

Our answer: the epidemiologic investigation and results of the investigation have been presented (Elbadawi LI, IDWeek, New Orleans LA, October 26-30, 2016, which is now cited; and will be published in more details elsewhere (see above comment). We are also citing the two following URLs:

<https://www.dhs.wisconsin.gov/disease/elizabethkingia.htm>

<https://www.cdc.gov/elizabethkingia/outbreaks/>

2. ICEEa1 element: the text is somewhat inconsistent. Line 306 states that ICEEa1 was unique to the outbreak isolates. Line 334 notes that "...it was present in all outbreak isolates but was absent in most others". Line 354 states that it was "only present in 4 isolates of the Singapore outbreak and 2 others." This actually accounts for a sizable number of previously published sequences (6/30; 20% in Figure S11 – is that the correct denominator?) unrelated to the current outbreak. What was the insertion site in those other isolates (line 231 states it was not present at that position in other isolates).

Our answer: Thank you for this comment; we have clarified lines 306 (now 310) ('highly associated') and line 354 ("Other than in the Wisconsin outbreak isolates, the ICEEa1 element was observed in only 6 strains"). We have also made clear that the location in the 6 other strains was distinct from *mutY* ("In the 6 other strains, the ICEEa1 element was inserted in genomic locations distant from *mutY* (**Supplementary Figure 10**)") and we now show the insertion site in the non-outbreak isolates on the revised Figure S10, which as a consequence now has two panels.

3. One of the intriguing features associated with ICEEa1 is the potential nucleotide mutation bias. However – what was the distribution of SNPs in the non-outbreak ICEEa1-positive isolates? The text does not clearly specify but it appears that these isolates did not exhibit this bias.

Our answer: The pattern is not there. This is what we expect, since it is not the presence of the ICEEa1 that affects the mutation spectrum but the interruption of *mutY*, which is not the location of integration in the other strains. Figure S10 and our changes in the text (see previous comment) now clarify this point.

4. Paragraph starting at line 334: interesting observations but without experimental validation of any of the proposed processes, this remains highly speculative.

Our answer: We agree it is speculative but feel it is important to cite these annotations in order to stimulate future research on the importance of the ICEEa1 element in *E. anophelis* pathogenesis. We have modified the paragraph to be more cautious on the possible functional implications of these observations.

5. This study does not provide any additional clarity on the epidemiological sources of the outbreak. While the data suggest that there was one most common recent ancestor ~1 year prior to the first cases the manuscript demonstrates at least 6 different clusters. This genotypic information should now be used to inform the epidemiological investigations and focus on identifying sources of the clusters rather than the total outbreak.

Our answer: Thank you for this suggestion. The genotypic data was used in real time to investigate potential sources of the outbreak, including looking for commonalities among patients within the genotypic clusters. No common source(s) among patients within the genotypic clusters were identified.

Minor comments:

1. Line 362 – 364: speculative; should move into the discussion

Our answer: we moved the sentence to the discussion.

2. Line 211 / 212: The German *E. coli* outbreak in 2011 was from a single source. There is no indication that this *E. anophelis* outbreak can be traced to direct exposure from a single source and as such the two cannot be directly compared.

Our answer: We have removed the sentence on the *E. coli* outbreak.

3. Paragraph line 298: please state the major classes of anti-microbial resistance observed in the isolates.

Our answer: thank you for the suggestion. We now provide antimicrobial susceptibility data (new section at the end of Results)

Reviewer #3 (Remarks to the Author):

The authors present a comprehensive and clear analysis of the unique *Elizabethkingia anophelis* (EA) outbreak recently reported in US Midwest. 69 EA isolates from the outbreak were sequenced and analysed in the context of 45 other available *Elizabethkingia* genomes. The authors' major conclusions include: the outbreak was caused by a single EA sublineage that resolved into six unique subclusters, little to no within-host genetic diversity was observed within patients, the EA outbreak sublineage accrues ~24 SNVs per year and was estimated to have arisen ~1yr before the outbreak, and that this high mutational rate is likely attributable to a truncated MutY common to all outbreak isolates. A variety of genomic features of the outbreak strain are also reported, from virulence genes to conjugative elements.

This was a tremendously well-written paper describing a very carefully conducted series of analyses. The expertise of this reviewer lies largely in the genomic correlates of epidemiological trends and events, so my few minor suggestions/comments will attend to that area, rather than the comparative genomics work.

Our answer: Thank you for this general appreciation!

1. Figure 2 makes the spatio-temporal trends in the outbreak quite clear, but it would be nice to additionally link that back to Figure 1. Between Figure 1 and 2, there's no labelling to indicate which letter belongs to which country. Perhaps a more informative node labelling scheme for Figure 1 would be to assign a single colour to geography and use saturation to indicate an isolate's distance from the bulk of cases, which seem to be in Milwaukee. All Milwaukee-area strains would be, say, 100% blue, with the saturation decreasing with increasing geographic distance from Milwaukee. This would help give some spatial context to the tree in Figure 1.

Our answer: Thank you for your suggestion. We have added the letter code of each county to Figure 2 inset in order to link the two figures. We feel a saturation color scale would make it very hard to locate the origin of the isolates on Figure 1.

2. The phylogenetic tree seems to suggest independent acquisition from some

reservoir; however, there is an interesting pair in CSID3000521200 & CSID3000516360, whose genomes are identical. Is there any epidemiological information on these cases?

Our answer: These 4 patients did not have any facilities nor did they have any other potential exposures in common. We looked in great detail at the closely related isolates pairwise and in subclusters and re-reviewed the data. Our close examination of these pairs did not result in detection of any unique epidemiologic linkages.

Similarly, CSID3000521202 and CSID3000516028 are near-identical, and both come from the same pair of counties as the previously mentioned pair. Would be interesting to see some commentary on this, as well as any available information on which cases within a given county were associated with specific facilities.

Our answer: Please see above answer, which holds also for this pair.

3. The authors note that the outbreak EA sublineage contains a number of antimicrobial resistance determinants; a brief comment on how these relate to the course of antibiotics used in treating these patients would be interesting for the clinically-oriented reader.

Our answer: *E. anophelis* isolates are typically resistant to multiple antimicrobial agents. We now provide the susceptibility data, which are similar to data from *E. anophelis* isolates causing infections in other world regions. An entire section was added to address this important point.

Subsequent to January 15, 2016, information that could be used to support decisions regarding empiric therapy was included in a memorandum circulated to clinicians and infection preventionists statewide, and subsequently results of antimicrobial susceptibility testing (MIC data) for 5 of the outbreak isolates were also widely circulated to clinicians and infection preventionists and also posted on the Wisconsin Division of Public Health website. We are currently examining the treatments used among all patients with laboratory confirmed sterile site infections and are finding that even with the availability of AST results, the empiric treatments selected were heterogeneous. Regarding the treatment of the patient with the quinolone resistant isolate: Initial treatment in the emergency department included ceftriaxone, meropenem, and vancomycin. Treatment during hospitalization included rifampin, piperacillin-tazobactam, and minocycline. Treatment post-discharge included minocycline and rifampin. The patient survived. Of note, for this patient a fluoroquinolone was not provided because the AST testing result of the patients' *E. anophelis* isolate showed it as being resistant.

4. Thank you for sharing the tree! It might also be helpful to include the BEAST XML file that was used to derive the time-labelled tree.

Our answer: We now provide the BEAST xml files used based on SNP data and on cgMLST data, as supplementary data.

5. In a similar sharing vein, given the geographical and temporal nature of the data, it would be well-suited for presentation on microreact.org!

Our answer: We recognize this is a valuable suggestion. A microreact project was created: <https://microreact.org/project/SyaeGCjvg> and this information was added to the data availability section.

6. One last data-sharing comment - it might be helpful to mirror the supplemental tables at a site such as FigShare, rather than just on a single Dropbox folder. Because the supplemental tables are only available at the Dropbox folder, if that folder disappears, so does a very critical set of data from this analysis.

Our answer: We have followed the reviewer suggestion and now provide the data in FigShare. The link has been updated.

7. Line 237: what is CTnDOT?

Our answer: ICEs were previously called Conjugative Transposons (CTn), and CTnDOT is one of the experimental models for such elements. Notably, its mechanisms of insertion and excision (discussed in the referenced publications) mediated by its integrase intDOT have been well studied. This has been clarified in the text as follows: “ICE*Ea1*'s integrase is 64% similar to the integrase of CTnDOT, a well studied ICE”.

Reviewer #1 (Remarks to the Author):

I am satisfied with the responses to my queries.

Reviewer #2 (Remarks to the Author):

All comments have been addressed.

Reviewer #3 (Remarks to the Author):

This reader was delighted to see the revised version of the manuscript, which nicely addresses the major suggestions of the various reviewers, and is happy to recommend the revision for publication.